# The Cultivar Effect on the Taste and Aroma Substances of Hakka Stir-Fried Green Tea from Guangdong

**DOI:** 10.3390/foods12102067

**Published:** 2023-05-20

**Authors:** Zihao Qiu, Jinmei Liao, Jiahao Chen, Peifen Chen, Binmei Sun, Ansheng Li, Yiyu Pan, Hongmei Liu, Peng Zheng, Shaoqun Liu

**Affiliations:** 1College of Horticulture, South China Agricultural University, Guangzhou 510642, China; scau20222018004@stu.scau.edu.cn (Z.Q.); ljm19127614165@stu.scau.edu.cn (J.L.); cjhtea@stu.scau.edu.cn (J.C.); 15994968387@163.com (P.C.); binmei@scau.edu.cn (B.S.); scau20213137137@stu.scau.edu.cn (H.L.); zhengp@scau.edu.cn (P.Z.); 2Meizhou Runqi Culture and Technology Development Co., Ltd., Meizhou 514000, China; yiyupan2023@163.com

**Keywords:** Hakka stir-fried green tea, HPLC, GC-MS, OAV, multivariate statistical analysis, tea cultivar

## Abstract

The flavor and quality of tea largely depends on the cultivar from which it is processed; however, the cultivar effect on the taste and aroma characteristics of Hakka stir-fried green tea (HSGT) has received little attention. High-performance liquid chromatography (HPLC), gas chromatography–mass spectrometry (GC-MS), and sensory evaluations were used to detect and predict the essential taste and aroma-contributing substances of HSGTs made from Huangdan (HD), Meizhan (MZ) and Qingliang Mountain (QL) cultivars. Orthogonal partial least squares data analysis (OPLS-DA) ranked four substances that putatively distinguished the tastes of the HSGTs, epigallocatechin gallate (EGCG) > theanine > epigallocatechin (EGC) > epicatechin gallate (ECG). Ten substances with variable importance in projections (VIPs) ≥ 1 and odor activation values (OAVs) ≥ 1 contributed to their overall aromas, with geranylacetone having the most significant effect on HD (OAV 1841), MZ (OAV 4402), and QL (OAV 1211). Additionally, sensory evaluations found that HD was relatively equivalent to QL in quality, and both were superior to MZ. HD had a distinct floral aroma, MZ had a distinct fried rice aroma, and QL had a balance of fried rice and fresh aromas. The results provide a theoretical framework for evaluating the cultivar effect on the quality of HSGT and put forward ideas for future HSGT cultivar development.

## 1. Introduction

Tea is one of the most popular beverages in the world. Its success as a beverage can be attributed to its excellent health benefits, unique and stimulating tastes, and pleasant aromas [1,2,3]. Hakka stir-fried green tea (HSGT) is a type of roasted green tea from Guangdong, China [4]. The unique “fried rice aroma” and “sweet taste” are popular in the local Chinese Hakka market, as well as in the markets of the overseas Hakka diaspora, tea connoisseurs, and emerging tea novices [5].

The taste and aroma of tea can differ significantly due to the selected cultivar, processing techniques, climatic environment, and post-harvest transportation conditions, with cultivar being one of the most impactful factors [6,7]. Theoretically, fresh leaves of any tea cultivar can be used for HSGT; however, in actual production, only certain cultivars are used due to their ability to develop its unique flavors [8]. Huangdan (HD, No. GS13008-1985, Chinese Crop Varieties Examining Committee, 1985) is an excellent tea variety native to China that is suitable for processing into green teas, including HSGT; it is loved by tea drinkers because of the highly aromatic teas it produces [9]. Meizhan (MZ, No. GS13004-1985, Chinese Crop Varieties Examining Committee, 1985) is also good for making HSGT and other green teas [10] due to its robust taste and high scores in sensory evaluations [11,12]. In addition, Qingliang Mountain tea (QL), a local Guangdong variety grown mainly in the Qingliang Mountains of Meizhou where there is a long history of making HSGT, produces the representative fried rice fragrance when properly processed. However, there are a lack of scientific data available on the taste and aroma characteristics of HSGTs processed from these and other varieties.

The flavor of tea depends on its chemical composition. For example, catechins, caffeine, and theanine provide astringent, bitter, and fresh tastes, respectively, and volatile compounds make up the abundant variety of tea aromas [13,14]. To date, more than 1400 different metabolites have been extracted and identified in tea plants. Although most of them are widely distributed in the plant kingdom, catechins, caffeine, theanine and a range of volatile chemicals are unique to tea in terms of their relative proportions [15,16,17]. Therefore, analyzing the main taste contributors and volatile compounds of HSGT produced from HD, MZ and QL to determine markers unique to high-quality HSGTs is worthwhile.

Among the various analytical methods used to characterize teas, high-performance liquid chromatography has been widely used to identify soluble compounds with high accuracy and sensitivity [18], whereas gas chromatography–mass spectrometry (GC-MS) has become one of the most common methods for the separation and identification of volatile compounds in tea [19,20,21]. The orthogonal partial-least square discriminant analysis (OPLS-DA) and the variable importance in projection (VIP > 1) value are commonly used to select potential characteristic chemical components from the large data sets generated from these analyses [22,23,24]. Non-volatile compounds must meet a certain threshold value to be perceptible to humans. This value, the odor activity value (OAV), is calculated as the ratio of the concentration of an individual volatile to its odor threshold value in water [25,26]. Volatiles with a VIP > 1 and an OAV ≥ 1 were selected as potential differential volatiles and were further evaluated for their contribution to the aroma [27,28].

In this study, the taste and aroma characteristics of three varieties, HD, MZ and QL, were fully investigated using data from HPLC, GC-MS, and sensory evaluations. It is expected that these data will provide a theoretical basis for determining the substances that characterize the distinctive tastes and aromas of HSGT from different cultivars.

## 2. Materials and Methods

### 2.1. Samples

Huang Dan stir-fried green tea (HD) and Meizhan stir-fried green tea (MZ) were produced by Jiexi Yidao Pharmaceutical Co., Ltd. (Jieyang, China), and Qingliang Mountain stir-fried green tea (QL) was produced by Runqi Cultural Technology Development Co., Ltd. (Meizhou, China). HSGT was produced according to a standard procedure (GB/T 14456.1-2008): each of the samples was subjected to withering, fixing, roasting, twisting, drying, and re-drying during the harvest season of September 2020 (Figure 1). The experiment was conducted with three biological replicates.

### 2.2. Chemicals

Caffeine standards were purchased from Beijing Weiye Research Institute of Metrology and Technology (Beijing, China). The reference standards, decanoic acid ethyl ester, theanine, catechin (C), epicatechin (EC), gallocatechin (GC), epigallocatechin (EGC), epicatechin gallate (ECG), gallocatechin gallate (GCG), and epigallocatechin gallate (EGCG), were purchased from Shanghai Yuanye Bio-Technology Co., Ltd. (Shanghai, China). Alkane standard solutions (C8–C40) for linear retention index (RI) calculations were obtained from TanMo Quality Testing Technology Co., Ltd. (Beijing, China). Internal standard solutions were prepared in dichloromethane prior to use. Ultrapure water was prepared with a Barnstead GenPure Pro system (Thermo Fisher Scientific, Waltham, MA, USA).

### 2.3. Analysis of Caffeine, Theanine and Catechins

Extracts of caffeine, theanine, and catechins were analyzed using HPLC (Waters Alliance 26,952,489 UV/Vis; Waters Technologies, Milford, MA, USA) and identified by comparing the retention times with authentic standards. Quantitative analyses of these compounds were conducted using calibration curves [29]. For analysis of caffeine, a 30 mL solution of 1.5% magnesium oxide was added to 0.1 g of freeze-dried tea powder and extracted in ultra-pure water (*w*/*v*) for 30 min at 100 °C. Then, 1 mL of each extract was filtered twice using 0.22 mm Millipore membranes. Ten μL of the filtrate was injected into an XSelect HSS C18 SB column (4.6 × 250 mm, 5 mm, Waters Technologies, Milford, MA, USA) at a flow rate of 0.9 mL/min and a column temperature of 35 ± 1 °C. The mobile phases consisted of 100% methanol (A) and 100% ultra-pure water (B). The compounds were separated isocratically at 30% A/70% B. Caffeine was detected at 280 nm.

Theanine was extracted from 0.1 g of freeze-dried tea powder in 10 mL of ultrapure water at 100 °C for 30 min. Next, 10 µL of the supernatant was filtered using a 0.22 mm Millipore membrane and injected into an RP-C18 column (250 mm × 4.0 mm, 5 μm, 35 ± 1 °C) at a flow rate of 0.5 mL/min. The specific HPLC procedure was based on the method of Mei et al. [30]. Detection was at 210 nm.

Catechins were extracted from 0.2 g of fine freeze-dried tea powder in 8 mL of 70% methanol. One mL of the supernatant was filtered through a 0.22 mm Millipore membrane followed by injection into an XSelect HSS C18 SB column (4.6 × 250 mm, 5 mm, Waters Technologies, Milford, MA, USA). The catechin monomer was eluted using a gradient elution procedure with 0.1% aqueous formic acid (*v/v*) (A) and 100% acetonitrile (B) as the mobile phases. The gradient elution started at 8% B for 5 min, was increased to 25% from 5–14 min, and decreased to 8% from 14–30 min. Detection was at 280 nm.

### 2.4. GC-MS Analysis of Volatile Compounds

Identification and quantification of volatiles were performed with headspace gas chromatography as in Chen et al. [31] using an Agilent 1890B gas chromatograph and a 5977A mass spectrometer (Agilent, Santa Clara, CA, USA) with an HP-5 MS column (30 m × 0.25 mm × 0.25 μm film thickness). A solid-phase microextraction device consisting of divinylbenzene/carboxylic acid/polydimethylsiloxane (DVB/CAR/PDMS) fibers (50/30 μm inner diameter, 2 cm length; Supelco, Darmstadt, Germany) was inserted into headspace vials and compounds were extracted at 80° for 40 min. After extraction, the SPME fibers were inserted into the GC-MS at 250 °C for 3 min.

The column flow rate was 1.0 mL/min with high-purity helium (purity ≥ 99.99%) as the carrier gas, and the solvent delay time was 4 min. The initial temperature was 50 °C for 1 min, and the temperature was ramped to 220 °C at 5 °C/min for 5 min. The temperature of the ion source was 230 °C and the electron impact (EI) ionization source was operated at 70 eV. The scan range was 30–400 amu.

Quantification was conducted using established methods [32]. Briefly, the peak area ratio of the internal standard and the target compound was calculated, and the actual concentration of the target compound was determined based on the ratio of their concentrations. Furthermore, the compounds were identified by cross-referencing the National Institute of Standards and Technology, (NIST, https://webbook.nist.gov/, accessed on 10 November 2022) mass spectrometry database with the compounds’ retention index (RI), which was determined using n-alkanes C9–C21. The RI was calculated as follows:RI = 100n + 100 [RT(x) − RT(n)]/[RT (n + 1) − RT(n)]
where RT(x) is the retention time of compound x and RT(n) and RT(n + 1) are the retention times of alkanes with carbon numbers n and n + 1 that immediately preceded and followed the elution of the compound [31].

### 2.5. Calculation of Odor Activity Values (OAVs)

The OAV is the ratio of the concentration of an aroma component to its odor threshold in an aqueous solution. The OAV was used to evaluate the contributions of each volatile compound to the aroma of the tea samples. The OAV of each volatile compound (OAV_i_) was calculated as OAV_i_ = C_i_/T_i_, where C_i_ is the concentration of compound i (μg kg^−1^) and T_i_ is the threshold value of compound i (μg kg^−1^).

### 2.6. Sensory Evaluation

The tea samples were evaluated and scored by four tea experts from South China Agricultural University with national senior tea assessor qualifications. All panel members had more than five years of experience in descriptive sensory analysis of tea. According to the standard method for sensory evaluation of tea (GB/T 23776-2018) [33], 150 mL of boiling water was added to 3 g of each tea sample in separate teacups with lids for 4 min to obtain tea infusions. Overall acceptability was given by each member of the group. Scores for taste and scent strength ranged from 0 to 10, with 0–2 signifying very weak intensity, 2–4 weak intensity, 4–6 neutral intensity, 6–8 strong intensity, and 8–10 extremely strong intensity [34]. Mean values were used to express the data.

### 2.7. Statistical Analysis

The original data from three replicates were processed using Microsoft Excel 2021. A one-way analysis of variance (ANOVA) with Tukey’s post hoc test was conducted using the SPSS 26 software package (SPSS Inc., Chicago, IL, USA) to analyze significant differences among the three samples. Radar diagrams were carried out using Origin 2022 statistical software (OriginLab Corporation, Northampton, MA, USA). Orthogonal partial least-squares discriminant Analysis (OPLS-DA) and variable importance in projection (VIP) were performed using SIMCA (Version14.1, Umetrics, Umea, Sweden) software. TBtools (Version 1.115, Guangzhou, China) was used to plot the heatmaps.

## 3. Results

### 3.1. Quantification of Caffeine, Catechins, and Theanine in HSGTs

Caffeine, catechins, and theanine were quantified in the HSGTs using HPLC (Figure 2A, Appendix A). The results showed that caffeine, ECG, and EGCG were significantly higher in HD than in MZ and QL, whereas MZ contained significantly higher levels of C, GC, and GCG compared with HD and QL (Appendix A). Moreover, the contents of EC, EGC, and theanine were significantly higher in QL than in HD and MZ. Six of the measured compounds, ECG, EGC, EGCG, GC, GCG, and theanine, differed significantly in all teas (Figure 2A), which suggested that they may have been the main non-volatile components shaping the taste differences among the three HSGTs.

A multivariate statistical analysis, OPLS-DA, which amplifies between-group differences and minimizes within-group differences, was used to visually simplify the HPLC results [35,36,37]. The score plot (Figure 2B) shows that the three HSGTs were clearly distinct in terms of their non-volatile profiles. The OPLS-DA permutation plots revealed that the model was reliable and was not overfitted (Figure 2C). The extent of the contribution that each variable has to an OPLS-DA model is expressed by the variable influence on projection (VIP), where those with VIP > 1 have a statistically significant contribution. The results showed that EGCG, theanine, EGC, and ECG, were the most likely causes of the differences in the taste of the three HSGTs (Figure 2D).

### 3.2. Volatile Components of HSGT

#### 3.2.1. Identification and Analysis of Volatiles

Using GC-MS, a total of 60 volatile compounds were identified in the three HSGTs. Among them, 40 were identified in HD, 31 in MZ, and 44 in QL. The volatile compounds included 10 alcohols, 4 aldehydes, 7 ketones, 17 alkenes, 11 esters, and 11 others (Appendix A). To discover more about the aroma of HSGTs, we analyzed the ratios of these 60 compounds (Appendix A). The alkenes were the highest in portion in HD, MZ, and QL (27%, 29% and 23%, respectively). Alcohols (20%) and esters (17%) were the next most abundant in HD, whereas esters (23%) and others (19%) were the next most abundant in MZ. QL had a more balanced proportion of others (23%), esters (18%), and alcohols (18%) than HD and MZ.

To investigate the differences in the volatile components of HD, MZ and QL, the 60 volatile components were quantified (Appendix A). In addition, base peak chromatography (BPC) using GC-MS was conducted. The BPC analysis revealed that all samples exhibited prominent signal detection by mass spectrometry, broad peak capacity, and remarkable separation efficiency (Figure 3). We also used cluster analysis to draw heatmaps and divided them into three main parts based on the major up-regulated volatiles in HSGT made from each variety (Appendix A). The compound with the highest concentration was dihydrolinalool (443.92 ± 33.06 µg/kg), followed by indole (404.5 ± 31.14 µg/kg) and nerolidol (346.78 ± 25.54 µg/kg), which together putatively provide a wood, fruity, and floral aroma. Dihydrolinalool had the third-highest concentration in MZ and the second highest in QL. β-Ionone, which has a floral, sweet aroma, was the most abundant volatile in both MZ (609.74 ± 2.78 µg/kg) and QL (222.42 ± 10.38 µg/kg). In addition, linalool, geraniol, jasmone, and geranylacetone, were abundant in HD, which coincided with the characteristic floral aroma of HD. In MZ, benzaldehyde, β-cyclocitral, 1,2-dihydro-1,1,6-trimethylnaphthalene, and other fruity and woody flavors were abundantly concentrated. Although QL had the greatest variety of aromas, the overall concentration of volatiles was not as high as those in HD and MZ, which may have affected the intensity of its aroma. The above results suggested that differences in the proportion and concentration of volatile components may have contributed to the different aromas of the three HSGTs.

#### 3.2.2. Identification and Analysis of the Characteristic Aromas of HSGT

To explore the characteristic volatile components of HD, MZ, and QL, an OPLS-DA model was constructed based on the identification of 60 volatile compounds (Figure 4A). The OPLS-DA permutation plots revealed that the model had suitable predictability and was not overfitted (Figure 4B). Fifteen volatiles had a VIP > 1, including three alcohols, two aldehydes, four ketones, two alkenes, two esters, and two others (Appendix A).

The contribution of a volatile to the overall aroma is related to both its concentration and its odor activity value (OAV), which is defined as the minimum concentration at which the compound has a perceptible odor [6]. Volatiles satisfying the conditions of VIP > 1 and OAV ≥ 1 were selected as potential differentiating volatiles [27]. A total of 10, including five floral substances, three woody substances, and one each of green and fruity substances, were identified as characteristic volatile components of HD, MZ, and QL (Table 1). HD possessed the largest number of these aromatic species, with the floral substances having the highest OAVs, whereas MZ had the highest OAVs among various floral and woody substances, including β-ionone (72.59), geranylacetone (4402.83), 1,2-dihydro-1,1,6-trimethylnaphthalene (36.68), and β-cyclocitral (37.24). In accordance with its weaker aroma, QL possessed the lowest OAVs for the characteristic aroma substances.

### 3.3. Sensory Qualities of the HSGTs

In terms of taste, the HD variety made the least bitter HSGT, that of QL was the least astringent, and both varieties made a final product with a pleasing aftertaste, which was in contrast to the strong bitter aftertaste of the MZ variety (Figure 5A). As for the aroma (Figure 5B), HD had a distinct floral aroma, MZ had a strong fried rice aroma, and QL was the most balanced: a refreshing aroma mixed with a fried rice aroma. According to the scoring results from the panel of tea experts (Figure 5C), HD and QL scored higher than MZ, but there was no statistically significant difference. This means that, theoretically, there is no clear ‘winner’ in taste and aroma between HD, QL and MZ, but based on the actual evaluation by the experts, HD and QL performed better overall (Appendix A).

### 3.4. Correlation Analysis of the Taste and Aroma Characteristics of HSGT with the Sensory Evaluation Results

To understand the relationships between non-volatile (theanine, catechins, caffeine) and volatile substances and the sensory evaluation results of the three HSGTs, correlation analysis was carried out based on the sensory evaluation, along with four key non-volatile compounds and 10 important volatile compounds (Figure 6A). Theanine and EGC were negatively correlated with astringency (*p* < 0.01); ECG and EGCG had significant negative correlations with bitterness (*p* < 0.05); and EGC and EGCG were significantly and positively correlated with a sweet aftertaste. As for the correlations with the characteristic aroma substances (Figure 6B), eight were significantly correlated with a roasted aroma, five with a floral aroma (*p* < 0.01), and all the aromas were significantly and negatively correlated with green aroma.

## 4. Discussion

### 4.1. EGCG, Theanine, EGC, and ECG Formed the Characteristic Tastes of the HSGTs

The taste of tea is an important indicator of its quality [41,42,43], and theanine, catechins, and caffeine are the most important contributors to the taste of tea [15,16,17]. Among the many factors that affect the levels of these compounds, the cultivar is the most important [10]. In this study, theanine, catechins, and caffeine were quantified by HPLC in HSGTs made from three tea varieties. The OPLS-DA model found significant differences in some of these substances, predicting that differences in cultivars had a strong influence on the teas’ tastes.

Sensory evaluation by a panel of tea experts confirmed that the three HSGTs had different combinations of flavors (Figure 5A), and tea flavor characteristics were closely related to the concentrations of the distinctive flavor components (Figure 6A). EGCG, theanine, EGC, and ECG were identified as the characteristic differential non-volatiles (VIP > 1; Figure 2D). We found that the content of EGC and EGCG in HD and QL with higher (but not statistically significant) sensory evaluation scores was higher than the EGC and EGCG in MZ. In fact, the content of the four characteristic taste substances was low in MZ, which may explain its poor taste, relative to the HSTGs made from HD and QL.

EGC and EGCG have been reported to be responsible for the bitter and astringent tastes of green tea [44,45], and the bitter and astringent taste provided by EGC the is strongest of the catechin monomers, followed by EGCG [46]. Interestingly, these three substances had a higher content in HD than MZ and QL, but HD was less bitter in taste. Furthermore, the correlation analysis indicated that ECG and EGCG were negatively correlated with bitterness. This is probably due to the higher caffeine content of HD, which has been shown to combine with catechins to reduce bitterness [16,46]. However, the uniquely high level of theanine and low level of ECG in QL may be the basis for its better taste [47,48]. The results suggested that the correct proportion of non-volatile substances may be more important than the absolute amount in forming a pleasant taste.

The correlation analysis based on sensory attributes and non-volatile metabolites showed that EGCG, theanine, EGC, and ECG were strongly correlated with the taste of the teas, which is consistent with the findings of previous studies [49,50]. It is noteworthy that caffeine, as a significant contributor to the briskness of tea [51], did not contribute significantly to the three HSGTs under the OPLS-DA discrimination model, possibly because its content did not vary significantly between the three HSGTs, or because it interacted with other components. Therefore, the study of the interactions between taste metabolites may be helpful for enhancing the flavor quality of tea.

### 4.2. Geranylacetone, β-Ionone, Indole, Nerolidol, and Dihydrolinalool Formed the Characteristic Aromas of HSGTs

When evaluating the quality of tea, aroma is an indispensable indicator [52]. The varying proportions of volatile compounds are the basis for the unique aromas of teas [53,54]. Previous studies have shown that there are differences in the aroma composition of HSGT made from different varieties [55]. In this study, a total of 60 volatile compounds (including alcohols, aldehydes, ketones, olefins, esters, and other compounds) were detected by GC-MS in the HSGTs made from three different tea varieties, the contents and proportions of which varied among the three HSGTs.

Odor substances with high OAV and VIP values contribute significantly to tea aroma [27,56]. Among the 60 detected volatiles, 10 were predicted to significantly affect the aroma (OAV ≥ 1 and VIP > 1). These substances can further the understanding of the origin of aroma differences between different tea varieties. The results will help to select the main aromatic substances and further optimize the tea-making process, thus producing tea products that are more in line with market demand.

In previous studies, these 10 substances have been shown to make an important contribution in green tea. For example, β-ionone is a common active odorant in green tea varieties and it is associated with the intensity of floral and sweet aromas [57,58]; however, it was negatively correlated with the floral scents in this study (Figure 6B). Jasmone, geranylacetone, and D-limonene are key odorants responsible for the chestnut-like aroma of green tea [59,60]. Unlike previous studies, this study found that geranylacetone had a very high OAV in all three types of HSGT, and the correlation analysis revealed that it was positively correlated with the roasted flavor (Figure 6B). Therefore, it may be the key HSGT aroma compound contributing to its distinctive aroma. Differences between the key components reported in this study and those previously reported may be due to differences in tea varieties and places of origin. In addition, the correlation analysis revealed the key aromatic substances responsible for the floral notes, as well as those responsible for the roasted aroma, the mainstay of the HSGT aroma. However, due to the small size of the sample set in this study, this conclusion needs to be further verified in a larger sample set containing more tea varieties.

## 5. Conclusions

In the present work, HPLC, GC-MS, and OAV analyses, combined with multivariate statistics were used to identify the key taste and aroma compounds in HSGTs made from three tea cultivars (HD, MZ, and QL). The tastes of the teas differed mainly due to their EGCG, theanine, EGC, and ECG contents, which had the most significant differences in concentration among the three HSGTs (VIP > 1). Ten (out of 60) compounds were predicted to form the aromas of the teas (VIP >1 and OAV ≥ 1). Among them, dihydrolinalool, geranylacetone, β-ionone, and indole were the dominant volatiles, responsible for the floral and woody aromas of the HSGTs. According to sensory evaluation and correlation analysis, HD and QL had better flavors than MZ, and EGCG, theanine, EGC, and ECG had the strongest contributions to the taste of the teas. Correlations between the detected compounds and the sensory evaluation confirmed that floral and roasted aromas were the main flavor categories determining HSGT quality, and the 10 characteristic aroma substances represented by geranylacetone made important contributions to the aroma of the tea. The results presented here provide a basis for understanding the unique compounds responsible for the special taste and aroma of HSGT and suggest potential directions for breeding more varieties that are suitable for making HSGT.

## Figures and Tables

**Figure 1 foods-12-02067-f001:**
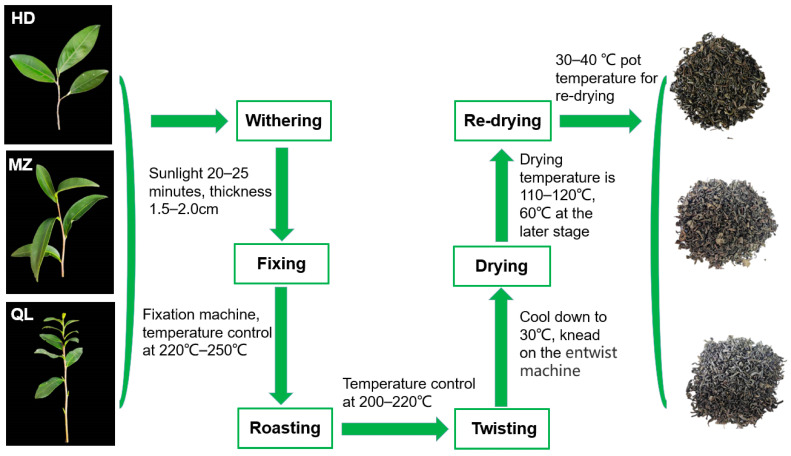
Tea varieties and processing steps.

**Figure 2 foods-12-02067-f002:**
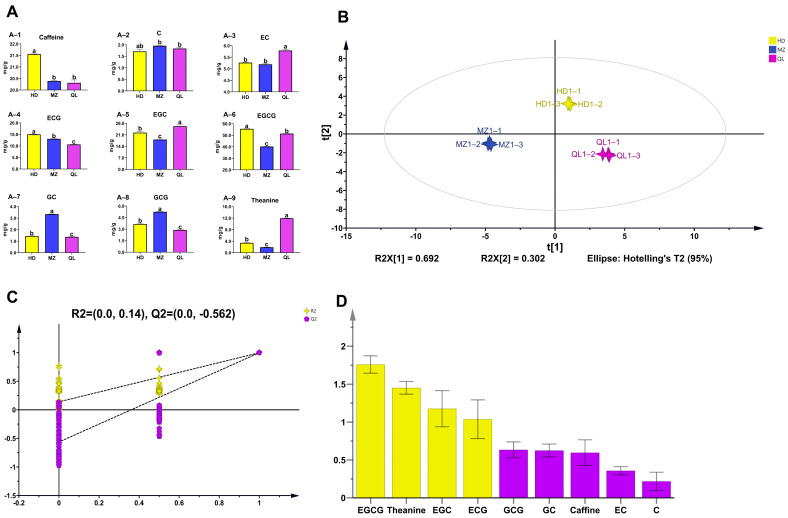
Multivariate statistical analysis of the non-volatile components in the three HSGTs. (**A**) The contents of theanine, catechin and caffeine were measured by high-performance liquid chromatography (HPLC). Different letters indicate statistically significant differences according to one-way ANOVA with Tukey’s post-hoc test; (**B**) OPLS-DA score plot; (**C**) cross-validation results: the intercept of the Q2 regression line of the cross-validation model with 200 tests of alignment was less than 0, indicating that the OPLS-DA discriminant model was not over-fitted and the model was relatively reliable. (**D**) VIP score plot, where yellow bars represent non-volatile compounds with VIP > 1 and purple bars represent those with VIP < 1.

**Figure 3 foods-12-02067-f003:**
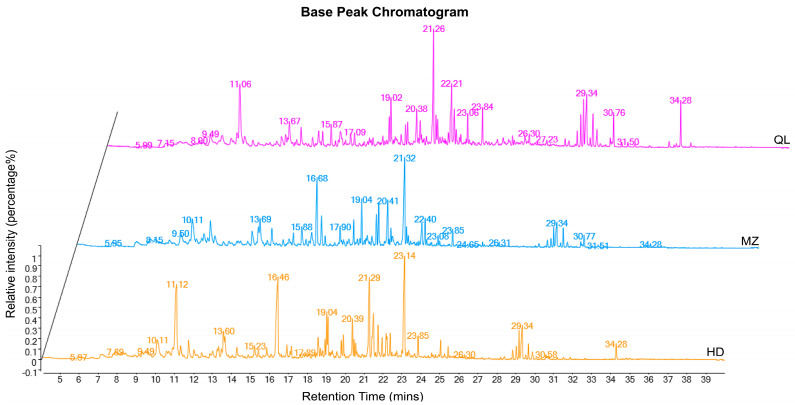
The GC-MS base peak chromatogram of volatile substances. The relative intensity is converted using the peak area normalization method.

**Figure 4 foods-12-02067-f004:**
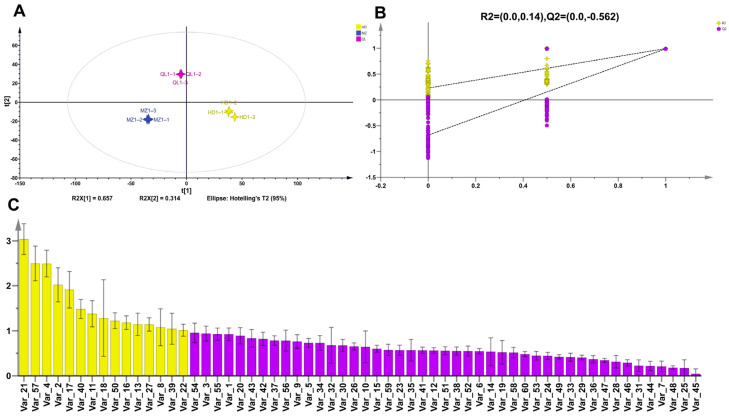
Multivariate statistical analysis of volatile components in the three HSGTs. (**A**) OPLS-DA score plot; (**B**) cross-validation results: the intercept of the Q2 regression line of the cross-validation model with 200 tests of alignment was less than 0, indicating that the OPLS-DA discriminant model was not over-fitted and the model was relatively reliable. (**C**) VIP score plot: yellow bars represent non-volatile compounds with VIP > 1; purple represents VIP < 1.

**Figure 5 foods-12-02067-f005:**
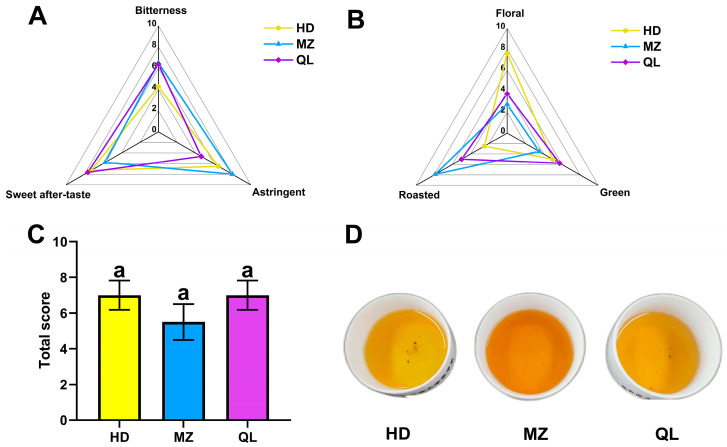
Sensory scores of (**A**) taste and (**B**) aroma; (**C**) overall acceptability scores, “a” refers to the significant difference between two levels calculated through One-Way ANOVA analysis; and (**D**) infusions of HSGTs.

**Figure 6 foods-12-02067-f006:**
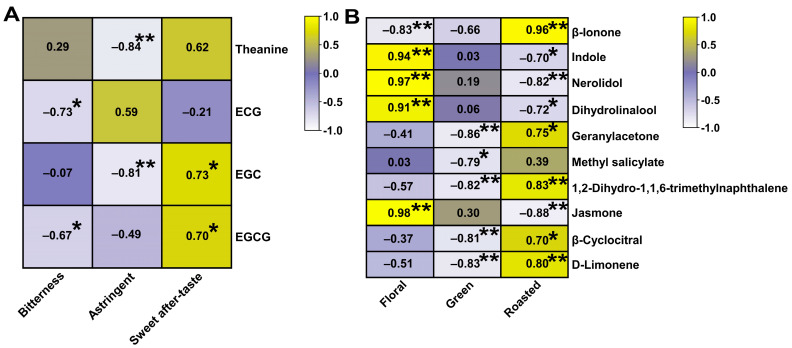
Correlation analysis of sensory evaluation of HSGTs with characteristics: (**A**) taste and (**B**) aroma compounds. Statistical significances are denoted by * for *p <* 0.05 and ** for *p <* 0.01.

**Table 1 foods-12-02067-t001:** The defining aromas of HSGTs.

Var. No.	Volatile Compounds	Odor Type	VIP	OT ^1^ (µg/L)	OAV		
					HD	MZ	QL
Var_21	β-Ionone	Floral	3.04	8.4 ^a^	0.00	72.59	26.48
Var_57	Indole	Floral	2.50	40 ^a^	10.11	3.10	1.24
Var_4	Nerolidol	Floral	2.49	250 ^a^	1.39	0.00	0.00
Var_2	Dihydrolinalool	Woody	2.02	70 ^c^	6.34	3.56	3.01
Var_17	Geranylacetone	Floral	1.91	0.06 ^b^	1841.00	4402.83	1211.00
Var_40	Methyl salicylate	Green	1.48	40 ^a^	1.91	2.92	0.27
Var_50	1,2-Dihydro-1,1,6-trimethylnaphthalene	Woody	1.22	2.5 ^a^	6.67	36.68	5.69
Var_16	Jasmone	Floral	1.18	7 ^a^	12.31	0.00	1.78
Var_13	β-Cyclocitral	Woody	1.14	3 ^a^	19.94	37.24	14.27
Var_22	D-Limonene	Fruity	1.01	34 ^a^	0.85	2.27	0.71

^1^ OT: odor threshold in water. ^a–c^: the threshold values of volatile compounds in water mentioned in the literature were sourced from the following references, with markers a–c sourced from the literature [38,39,40].

## Data Availability

Data is contained within the article.

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
