# Peer review of "The Cultivar Effect on the Taste and Aroma Substances of Hakka Stir-Fried Green Tea from Guangdong"

_foods, 2023, doi:10.3390/foods12102067_

Round 1

Reviewer 1 Report

The Cultivar Effect on the Taste and Aroma Substances of Hakka Stir-fried Green Tea from Guangdong

Reviewer comments

In this work, the authors apply chromatographic techniques, sensorial analysis and chemometrics analysis to carry out a study on the taste and aroma characteristics of three varieties of Hakka stir-fried green tea from Guangdong. Their aim is determining the substances that characterize the distinctive tastes and aromas of Hakka stir-fried green tea from different cultivars. The experimental data seem robust, correctly obtained, and adequate for the objective pursued. Furthermore, the results could help to obtain new Hakka stir-fried green tea varieties with organoleptic characteristics that meet market demand. However, I think there are some flaws in the paper that need to be fixed, as well as clarifying certain explanations or arguments.

Therefore, this reviewer considers the manuscript acceptable for publication in Foods after revision, specifically the following issues should be addressed:

- internal standard should be included together with the rest of products, in the 2.2 Chemical section.

- the one-way ANOVA table referred in section 3.1 should be included as Supplementary information

- I have not been able to find information in the document regarding the number of samples analyzed for each type of tea, or the number of replicates performed. This information is crucial for the statistical analysis results and should be included.

- the quantification process followed for LC and GC should be explained, especially regarding the calibration procedures carried out, concentration ranges, etc. The authors refer to reference 31 in the bibliography for this information, but I couldn't find it there.

- the research should include all the data used for chemometric analysis (otherwise, the quality of the results cannot be assessed), not just those corresponding to the volatile fraction. Therefore, tables with information on the quantification of the non-volatile fraction (Section 3.1) and sensory analysis should be included. These tables, along with Table 1, could be included as Supplementary Files.

- the authors do not indicate whether any analytical blanks were performed, which is particularly important for volatile analysis as it allows for the exclusion of possible contaminants or artifacts. For example, dibutyl phthalate is a ubiquitous contaminant, but it is listed among the detected volatile compounds.

- I believe Figure 3 is entirely unnecessary, since the information presented in Figure 3A is already detailed in the text, and nothing is mentioned about Figures 3B-D in the document.

- it could be very illustrative to include a figure with a representative GC-MS chromatogram of the volatiles obtained for each tea variety. This would provide a clear and intuitive understanding of their differences and the quality of the analyses.

- lines 242-243: "...we constructed an OPLS-DA model (Figure 4A)." Please, specify which data sets were used to construct the model.

- lines 284-285: "...Pearson correlation analysis was conducted." Please, clarify which datasets were correlated.

- Figure 6: please, specify the meaning of the asterisks.

Author Response

Dear Reviewer,

Thank you for your valuable comments on our paper. We appreciate your time and effort in reviewing our work and have carefully considered your suggestions. We have revised the text according to your recommendations and made sure that all the changes are consistent with the overall theme of the paper.

We hope that our revised version meets your expectations and would appreciate it if you could confirm that the corrections have been successful. If there are any further issues, please do not hesitate to let us know.

Once again, thank you for your help and guidance in improving our manuscript.

Best regards,

Peng Zheng

Reviewer 2 Report

Overall it is a well written and designed study on the good sensory properties of Hakka stir-fried green tea. The experimental work is sound with the  data clearly presented and discussed.This work opens up future studies for maximizing its utilization.

Author Response

Dear Reviewer,

I hope this email finds you well. I am writing to express our gratitude for your review of our paper on the sensory properties of Hakka stir-fried green tea. We were pleased to read your positive comments and feedback on our work. Your acknowledgement of the sound experimental work and clear presentation and discussion of the data was greatly appreciated. We are glad to know that you found our study to be well-written and well-designed. Moreover, we agree with your observation that this study opens up future opportunities for optimizing the utilization of Hakka stir-fried green tea. Once again, thank you for your time and valuable feedback. Your suggestions and critiques have helped us to improve the quality of our research. We look forward to hearing from you regarding the acceptance of our manuscript.

Best regards,

Peng Zheng
